# DP-SSL: Towards Robust Semi-supervised Learning with A Few Labeled Samples

**Yi Xu**[1]    **Jiandong Ding**[2]    **Lu Zhang**[1]    **Shuigeng Zhou** [1]*

[1]Shanghai Key Lab of Intelligent Information Processing,
and School of Computer Science, Fudan University, China
[2]Alibaba Group
{yxu17, jdding, l_zhang19, sgzhou}@fudan.edu.cn

## Abstract

The scarcity of labeled data is a critical obstacle to deep learning. Semi-supervised learning (SSL) provides a promising way to leverage unlabeled data by pseudo labels. However, when the size of labeled data is very small (say a few labeled samples per class), SSL performs poorly and unstably, possibly due to the low quality of learned pseudo labels. In this paper, we propose a new SSL method called DP-SSL that adopts an innovative data programming (DP) scheme to generate probabilistic labels for unlabeled data. Different from existing DP methods that rely on human experts to provide initial labeling functions (LFs), we develop a multiple-choice learning (MCL) based approach to automatically generate LFs from scratch in SSL style. With the noisy labels produced by the LFs, we design a label model to resolve the conflict and overlap among the noisy labels, and finally infer probabilistic labels for unlabeled samples. Extensive experiments on four standard SSL benchmarks show that DP-SSL can provide reliable labels for unlabeled data and achieve better classification performance on test sets than existing SSL methods, especially when only a small number of labeled samples are available. Concretely, for CIFAR-10 with only 40 labeled samples, DP-SSL achieves 93.82% annotation accuracy on unlabeled data and 93.46% classification accuracy on test data, which are higher than the SOTA results.

## 1   Introduction

The de-facto approaches to deep learning achieve phenomenal success with the release of huge labeled datasets. However, large manually-labeled datasets are time-consuming and expensive to acquire, especially when expert labelers are required. Nowadays, many techniques are proposed to alleviate the burden of manual labeling and help to train models from scratch, such as active learning [1], crowd-labeling [2], distant supervision [3], semi [4]/weak [5]/self-supervision [6]. Among them, semi-supervised learning (SSL) is one of the most popular techniques to cope with the scarcity of labeled data. Two major strategies of SSL are pseudo labels [7] and consistency regularization [8]. Pseudo labels (also called self-training [9]) utilize a model's predictions as the labels to train the model again, while consistency of regularization forces a model to make the same prediction under different transformations. However, when the size of labeled data is small, SSL performance degrades drastically in both accuracy and robustness. Fig. 1 shows the change of prediction error rate with the number of labeled samples of CIFAR-10. When the number of labeled samples reduces from 250 to 40, error rates of major existing SSL methods increase from 4.74% (USADTM) to 36.49% (MixMatch). One possible reason of performance deterioration is the quality degradation of learnt pseudo labels when labeled data size is small. Therefore, in this paper we address this problem by

---

*Corresponding author.

35th Conference on Neural Information Processing Systems (NeurIPS 2021).

developing sophisticated labeling techniques for unlabeled data to boost SSL even when the number of labeled samples is very small (e.g. a few labeled samples per class).

Recently, *data programming* (DP) was proposed as a new paradigm of weak supervision [10]. In DP, human experts are required to transform the decision-making process into a series of small functions (called *labeling functions*, abbreviated as LFs), thus data can be labeled programmatically. Besides, a label model is applied to determining the correct labels based on consensus from the noisy and conflicting labels assigned by the LFs. Such a paradigm achieves considerable success in NLP tasks [11–14]. In addition, DP has also been applied to computer vision tasks [15, 16]. However, current DP methods require human experts to provide initial LFs, which is time-consuming and expensive, and it is not easy to guarantee the quality of LFs. Furthermore, LFs specifically defined for one task usually cannot be re-used for other tasks.

In this paper, we propose a new SSL method called DP-SSL that is effective and robust even when the number of labeled samples is very small. In DP-SSL, an innovative data programming (DP) scheme is developed to generate probabilistic labels for unlabeled data. Different from existing DP methods, we develop a *multiple-choice learning* (MCL) based approach to automatically generate

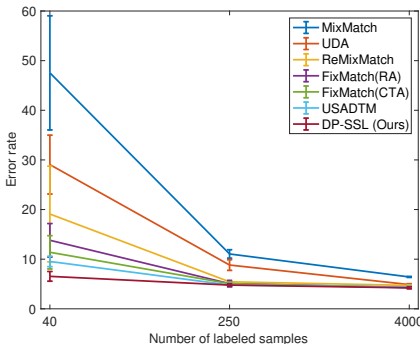

Figure 1: Error rate *vs*. #labeled samples (CIFAR-10). Results of existing methods are from the original papers. When only 40 labeled samples are given, all existing SSL methods are substantially degraded and more unstably, while our method is still effective and robust.

LFs from scratch in SSL style. To remedy the over-confidence problem with existing MCL methods, we assign an additional option as *abstention* for each LF. After that, we design a label model to resolve the conflict and overlap among the noisy labels generated by LFs, and infer a probabilistic label for each unlabeled sample. Finally, the probabilistic labels are used to train the end model for classifying unlabeled data. Our experiments validate the effectiveness and advantage of DP-SSL. As shown in Fig. 1, DP-SSL performs best, and only 1.76% increase of error rate when the number of labeled samples decreases from 250 to 40 in CIFAR-10.

Note that the pseudo labels used in existing SSL methods are quite different from the probabilistic labels in DP-SSL, which may explain the advantage of DP-SSL over existing SSL methods. On the one hand, pseudo labels are "hard" labels that indicate an unlabeled sample belonging to a certain class or not, while probabilistic labels are "soft" labels that indicate the class distributions of unlabeled samples. Obviously, the latter should be more flexible and robust. On the other hand, pseudo labels are actually generated by a single model for all unlabeled samples, while probabilistic labels are generated from a number of diverse and specialized LFs (due to the MCL mechanism), which makes the latter more powerful in generalization as a whole.

In summary, the contributions of this paper are as follows: 1) We propose a new SSL method DP-SSL that employs an innovative data programming method to generate probabilistic labels for unlabeled data, which makes DP-SSL effective and robust even when there are only a few labeled samples per class. 2) We develop a multiple choice learning based approach to automatically generate diverse and specialized LFs from scratch for unlabeled data in SSL manner. 3) We design a label model with a novel potential and an unsupervised quality guidance regularizer to infer probabilistic labels from the noisy labels generated by LFs. 4) We conduct extensive experiments on four standard benchmarks, which show that DP-SSL outperforms the state-of-the-art methods, especially when only a small number of labeled samples are available, DP-SSL is still effective and robust.

## 2 Related Work

Here we briefly review the latest advances in multiple choice learning, semi-supervised learning, and data programming, which are related to our work. Detailed information is available in [17–21].

## 2.1 Multiple Choice Learning

Multiple choice learning (MCL) [22] was proposed to overcome the low diversity problem of models trained independently in ensemble learning. For example, stochastic multiple choice learning [23] is for training diverse deep ensemble models. However, a crucial problem with MCL is that each model tends to be overconfident, which results in poor final prediction. To solve this problem, [24] forces the predictions of non-specialized models to meet a uniform distribution, so that the final decision is summed over diverse outputs. [25] proposes an additional network to estimate the weight of each specialist's output. In this paper, we develop an improved MCL based scheme to automatically generate diverse and specialized labeling functions (LFs) from scratch in an SSL manner. These LFs are used to generate preliminary (usually noisy) labels for unlabeled data.

## 2.2 Semi-supervised Learning

Semi-supervised learning (SSL) has been extensively studied in image classification [26], object detection [27], and semantic segmentation [28]. Two popular SSL strategies for image classification are pseudo labels [7] and consistency regularization [8]. Pseudo-label methods generate artificial labels for some unlabeled images and then train the model with these artificial labels, while consistency regularization tries to obtain an artificial distribution/label and applied it as a supervision signal with other augmentations/views. These two strategies have been adopted by a number of recent SSL works [4, 8, 29–39]. For example, FixMatch [4] proposes a simple combination of pseudo labels and consistency regularization. [36] employs unsupervised learning and clustering to determine the pseudo labels. In this paper, we propose a new SSL method that is effective and robust even when the size of labeled data is very small. Our method employs an innovative data programming alike method to automatically generate probabilistic labels for unlabeled data.

## 2.3 Data Programming

Data programming [10] is a weak supervision paradigm proposed to infer correct labels based on the consensus among noisy labels from labeling functions (LFs), which are modules embedded with decision-making processes for generating labels programmatically. Following the DP paradigm, Snorkel [12] and Snuba [40] were proposed as rapid training data creation systems. Their LFs are built with various weak supervision sources, like pattern regexes, heuristics, and external knowledge base etc. Recently, more works are reported in the literature [11, 13–16, 21, 41–48]. Among them, [11, 13, 14, 45–47] focus on the adaption of label model in DP. For example, [21] aims to reduce the computational cost and proposes a closed-formed solution for training the label model. [15, 16, 41–43] apply DP to computer vision. Concretely, [16, 42, 43] heavily rely on the pretrained models. [41] combines crowdsourcing, data augmentation, and DP to create weak labels for image classification. [15] presents a novel view for resolving infrequent data in scene graph prediction training datasets via image-agnostic features in LFs. However, all these methods cannot be directly applied to training models from scratch with a small number of labeled samples. Thus, in this paper we extend DP by exploring both MCL and SSL to generate arbitrary labeling functions.

# 3 Method

For a $C$-class SSL classification problem, assume that all training data $X$ are divided into labeled data $X_l$ and unlabeled $X_u$, and test data are denoted as $X_t$. Following the notation in [4, 36], $\{x_l, x_l^w\} \in X_l$ are the paired labeled samples with labels $y_l \in \{1, \ldots, C\}$, and $\{x_u, x_u^w, x_u^s\} \in X_u$ are the triple unlabeled samples. Here, $x_l$ and $x_u$ represent the raw images without any transformations. $x_l^w, x_u^w$, and $x_u^s$ are the images based on the weak and strong augmentation strategies, respectively. In this paper, weak augmentation uses a standard flip-and-shift strategy, and strong augmentation uses the RandAugment [49] strategy with Cutout [50] augmentation operation.

## 3.1 Background

**FixMatch**. In the FixMatch [4] algorithm, apart from basic cross-entropy on labeled samples, consistency regularization with pseudo labels on unlabeled samples is represented as:

$$\mathcal{L}^{FM}(x_u^w, x_u^s) = \mathbb{1}(max(p(y|x_u^w)) \geq \epsilon) \cdot H(\hat{y}_u^w, p(y|x_u^s)), \tag{1}$$

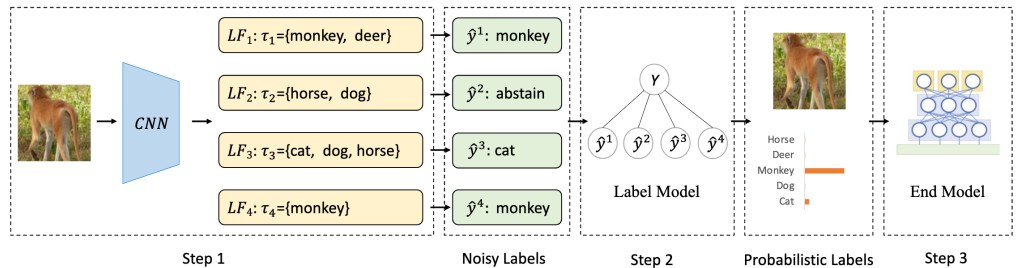

Figure 2: Framework of the DP-SSL method with four LFs.

where $H$ is the cross-entropy, $\tau$ is the pre-defined threshold, $p$ represents the output probability of the model, and $\hat{y}_u^w := \arg max(p(y|x_u^w))$ is the pseudo label from the weakly augmented predictions.

**Multiple Choice Learning**. Stochastic Multiple Choice Learning (sMCL) [23] aims to specialize each individual model on a subset of data, by minimizing the loss as follows:

$$\mathcal{L}^{sMCL}(x_l, y_l) = \min_{k \in \{1, \cdots, K\}} H(y_l, p_k(y|x_l)), \tag{2}$$

where $p_k$ is the output probability of the $k$-th model. For a training sample $(x_l, y_l)$, sMCL feeds the data to all $K$ models but only chooses the most accurate model to do back-propagation. Consequently, each model performs better on some classes than the other models, *i.e.*, each model becomes a specialist on some particular classes.

### 3.2 Framework

Fig. 2 shows the framework of our DP-SSL method, which works in three major steps as follows:

- Step 1. We employ an MCL based approach to automatically generate $K$ LFs from scratch in an SSL style. Here, each LF is trained on a subset of $C$ classes in the training set based on MCL. As shown in Fig 2, the 2nd LF is trained with samples of classes "horse" and "dog", and abstains from predicting when facing monkey images.

- Step 2. A graphical model is developed as the label model to aggregate the noisy labels and produce probabilistic labels for unlabeled training data. The label model is learned in an SSL manner with an additional regularizer.

- Step 3. The end model is trained with both provided labels and probabilistic labels generated from Step 2. Finally, we verify the performance of the end model on the test data.

### 3.3 Labeling Function

In Step 1 of our method, LFs are exploited to generate noisy labels for each unlabeled image. In previous DP works for computer vision, LFs are built via external image-agnostic knowledge [15] or pretrained models [16, 42, 43]. However, it is difficult to explicitly describe the rules of image classification. Instead, here we innovatively explore MCL and SSL for automatic LF generation.

As shown in Fig. 2, we share the same backbone (Wide ResNet [51] in this paper) to extract features of images for multiple prediction heads (called LFs in this paper). To promote the diversity of LFs, we transform the features and feed each LF with different transformed features as follows:

$$f_k = \sum_{j=1}^{HW} \frac{e^{-\beta_k dis(f[j], c_k)}}{\sum_{k'=1}^{K} e^{-\beta_{k'} dis(f[j], c_{k'})}} (f[j] - c_k). \tag{3}$$

In this paper, $K$ is the number of LFs, $f \in \mathbb{R}^{HW \times D}$ denotes the feature map of input image $x$ before global average pooling of backbone, $f[j] \in \mathbb{R}^D$ is the feature vector at the spatial position $j$ of $f$. $c_k \in \mathbb{R}^D$ is the learnable clustering center of the $k$-th LF, $\beta_k$ is the learnable variable of the $k$-th cluster, $dis(A, B)$ represents the distance between $A$ and $B$. Thus, $f_k$ corresponds to the feature fed

to the $k$-th LF, and describes the $k$-th aggregated pattern of $f$ among the $K$ centers, it can also be considered as a learnable weighted spatial pooling for feature $f$. Then, supposing $\mathcal{F}_k$ is the classifier in the $k$-th LF, which would output the probability $\mathcal{F}_k(f_k)$ as prediction. For clarity, in the following we denote $p_k(y|x) := \mathcal{F}_k(f_k)$ of the $k$-th LF with the input image $x$.

As depicted in [23], the classifiers lack diversity of prediction even trained with different protocols. Therefore, we adopt MCL to assign a subset of labeled data for each classifier automatically to improve diversity. However, it is intuitive to observe that in Eq. (2) each category can only be assigned to one LF, and no consensus can be exploited. Therefore, we increase the proportion of selected models in Eq. (2) to do back-propagation, which is formulated as

$$\mathcal{L}_l^{MCL}(x_l^w, y_l) = \min_{\substack{\mathcal{M} \subset \{H(y_l, p_k(y|x_l^w))\}_{k=1}^K \\ |\mathcal{M}| = \rho \cdot K}} \frac{1}{\rho \cdot K} \sum_{k'=1}^{\rho \cdot K} \mathcal{M}_{k'}, \tag{4}$$

where $\mathcal{M}_{k'}$ indicates the $k'$-th element in the set $\mathcal{M}$, and $\rho \in [1/K, 1]$ is a designed parameter to represent the ratio of specialist LFs. When $\lfloor \rho \cdot K \rfloor$ is equal to 1, Eq. (4) becomes the traditional MCL in Eq. (2). In contrast, if $\lfloor \rho \cdot K \rfloor$ is equal to K, it deteriorates to the basic ensemble learning, where all $K$ classifiers are trained with the same data.

Based on MCL, each LF is a specialist for some classes, so it can get high accuracy for samples in these classes. While for samples from other classes not specialized by the LF, it fails to predict due to over-confidence. Thus, we take only the probabilities of specialized categories as predictions, and allow each LF to abstain from some samples in the dataset. Formally, we denote '0' as the abstention label, and the specialized category set of the $k$-th LF as $\tau_k = \{\tau_k^1, \ldots, \tau_k^{|\tau_k|}\}$. Then, the output label of the $k$-th LF $\hat{y}^k$ satisfies $\hat{y}^k \in \{0\} \cup \tau_k$, e.g., the output of the 1st LF in Fig. 2 is among "monkey", "deer" and "abstention" because its specialized category set $\tau_1 = \{\text{monkey}, \text{deer}\}$. Then, we denote the probability over the specialized category set $\tau_k$ and "abstention" option as $\bar{p}_k(y|x)$, where $\bar{p}_k(y|x) \in \mathbb{R}^{|\tau_k|+1}$. The objective function over labeled samples with abstention option is

$$\mathcal{L}_l(x_l^w, y_l) = \sum_{k=1}^K \big( \mathbb{1}(y_l \in \tau_k) H(y_l, \bar{p}_k(y|x_l^w)) + \mathbb{1}(y_l \notin \tau_k) H(0, \bar{p}_k(y|x_l^w)) \big), \tag{5}$$

Then, for the unlabeled training data, we follow the settings in FixMatch [4], where unlabeled data are supervised by the pseudo labels $\hat{y}_u^{w,k}$ of weak augmentation data $x_u^w$. Thus,

$$\mathcal{L}_u(x_u^w, x_u^s) = \sum_{k=1}^K \mathbb{1}(\max(\bar{p}_k(y|x_u^w)) \geq \epsilon) \Big( \mathbb{1}(\hat{y}_u^{w,k} \in \tau_k) H(\hat{y}_u^{w,k}, \bar{p}_k(y|x_u^s)) +$$

$$\mathbb{1}(\hat{y}_u^{w,k} \notin \tau_k) H(0, \bar{p}_k(y|x_u^w)) \Big), \tag{6}$$

where $\hat{y}_u^{w,k} := arg\max(\bar{p}_k(y|x_u^w))$. Specifically, we only keep samples whose largest probability (including the abstention option) is above the predefined threshold $\epsilon$ (0.95 in our paper), and train the model on the kept data with pseudo label $\hat{y}_u^{w,k}$. Accordingly, the training in this step is to minimize the objective function as follows:

$$\mathcal{L}(x_l^w, y_l, x_u^w, x_u^s) = \mu_l^{MCL} \mathcal{L}_l^{MCL}(x_l^w, y_l) + \mu_l \mathcal{L}_l(x_l^w, y_l) + \mu_u \mathcal{L}_u(x_u^w, x_u^s), \tag{7}$$

where $\mu_l^{MCL}$, $\mu_l$ and $\mu_u$ are hyper-parameters. In our implementation, we first set $\mu_l^{MCL} = 1$ and $\mu_l = \mu_u = 0$, then adjust $\mu_l^{MCL}$ to 0 and $\mu_l = \mu_u = 1$ after the convergence of $\mathcal{L}_l^{MCL}$.

Generally, in Step 1, MCL is expected to generate specialized class sets $\tau$ for LFs, with which samples are more easily discriminated by SSL classifiers even there are a few labeled samples. Besides, the abstention option is for addressing the over-confidence problem of samples from non-specialized sets.

### 3.4 Label Model

In Step 2 of our method, we utilize a graphical model to specify a single prediction by integrating noisy labels provided by $K$ LFs. For simplification, we assume that the $K$ LFs are independent (as shown in Fig. 2). Then, suppose that $\hat{\mathbf{y}} = (\hat{y}^1, \cdots, \hat{y}^K)^\top \in \mathbb{R}^K$ is the vectorized form of the

predictions from $K$ LFs, the joint distribution of the label model can be described as:

$$P(y, \hat{\mathbf{y}}) = \frac{1}{Z} \prod_{k=1}^{K} \phi(y, \hat{y}^k) \tag{8}$$

where $Z$ is the normalizer of the joint distribution, $\phi$ is the potential that couples the target $y$ and noisy label $\hat{y}^k$. In this paper, we extend the dimension of parameters $\theta$ in label model to $K \times C$ to support multi-class classification. Set $e_{ky} := exp(\theta_{ky})$, which is the exponent of parameters $\theta_{ky}$. Now we are to construct the potential function $\phi$. Due to the specialized LFs, the potential $\phi$ should benefit the final prediction when a noisy label agrees with the target. That is, we should have $\phi(y, \hat{y}^k) > 1$. Thus, we set $\phi$ as $1 + e_{ky}$ for this case. On the contrary, the potential $\phi$ should negatively impact the final prediction when a noisy label conflicts with the target label in the specialized category set, i.e., we should have $\phi(y, \hat{y}^k) < 1$. Therefore, for this case we set $\phi$ to $1/(1 + e_{ky})$. For the other cases, we follow the design in [52]. In summary, the potential $\phi$ is defined as follows:

$$\phi(y, \hat{y}^k) = \begin{cases} 1 + e_{ky}, & if \ y \in \tau_k, \ \hat{y}^k \in \tau_k, \ \hat{y}^k = y \\ 1/(1 + e_{ky}), & if \ y \in \tau_k, \ \hat{y}^k \in \tau_k, \ \hat{y}^k \neq y \\ e_{ky}, & if \ y \notin \tau_k, \ \hat{y}^k \in \tau_k, \ \hat{y}^k \neq y \\ 1. & otherwise \end{cases} \tag{9}$$

With the potential above, the normalizer $Z$ of the joint distribution in Eq. (8) can be obtained by summarizing over $y$ and $\hat{y}^k$:

$$\begin{aligned} Z &= \sum_{y \in \mathcal{Y}} \prod_{k=1}^{K} \sum_{\hat{y}^k \in \{0\} \cup \tau_k} \phi(y, \hat{y}^k) \\ &= \sum_{y \in \mathcal{Y}} \prod_{k=1}^{K} \left( \mathbb{1}(y \in \tau_k)(2 + e_{ky} + \frac{|\tau_k| - 1}{1 + e_{ky}}) + \mathbb{1}(y \notin \tau_k)(1 + |\tau_k| e_{ky}) \right). \end{aligned} \tag{10}$$

Then, the objective function of the label model can be expressed in an SSL manner as follows:

$$\mathcal{L}(\hat{\mathbf{y}}_l, y_l, \hat{\mathbf{y}}_u) = \underbrace{\sum_{x_l} H(y_l, P(y, \hat{\mathbf{y}}_l))}_{\text{labeled samples}} + \underbrace{(-\sum_{x_u} \log \sum_{y \in \mathcal{Y}} P(y, \hat{\mathbf{y}}_u))}_{\text{unlabeled samples}} + R(\theta, \hat{\mathbf{y}}_u), \tag{11}$$

where the first part is the cross-entropy loss, the second is the negative log marginal likelihood on the observed noisy labels $\hat{\mathbf{y}}_u$, and the third is a regularizer. In our method, the regularizer is utilized to guide the label model with statistical information (the accuracy of each LF). However, the accuracy of each LF on noisy labels is unavailable, while the accuracy on labeled training is almost 100% due to over-fitting. Thus, we have to estimate the accuracy of each LF with the observable noisy labels $\hat{\mathbf{y}}$, which will be presented in Sec. 3.5. After training, the label model produces probabilistic labels $\pi$ by computing the joint distribution in Eq. (8) with the noisy labels $\hat{\mathbf{y}}$.

### 3.5 Accuracy Estimation

Now, we formally describe our method for estimating the accuracy of LFs. We transform the multi-class problem into $C$ one-versus-all tasks. For the $i$-th ($i \in [1, \cdots, C]$) one-versus-all task, we denote the unobserved ground-truth labels as $z_i \in \{\pm 1\}$ ($z_i = +1$ means $y = i$, and $z_i = -1$ represents $y \neq i$), noisy labels of the $k$-th LF as $\hat{z}_i^k \in \{\pm 1, 0\}$,

$$\hat{z}_i^k = \begin{cases} 1 & if \ \hat{y}^k = i, \\ 0 & if \ \hat{y}^k = 0, \\ -1 & otherwise. \end{cases} \tag{12}$$

Then, we can write $\mathbb{E}[\hat{z}_i^k z_i]$ as

$$\begin{aligned} \mathbb{E}[\hat{z}_i^k z_i] &= P(\hat{z}_i^k z_i = 1) - P(\hat{z}_i^k z_i = -1) \\ &= P(\hat{z}_i^k z_i = 1) - (1 - P(\hat{z}_i^k z_i = 1) - P(\hat{z}_i^k z_i = 0)) \\ &= 2P(\hat{z}_i^k = z_i) + P(\hat{z}_i^k = 0) - 1. \end{aligned} \tag{13}$$

Assume that $\hat{z}_i^j \perp\!\!\!\perp \hat{z}_i^k | z_i$ for distinct $j$ and $k$, then

$$\mathbb{E}[\hat{z}_i^j \hat{z}_i^k] = \mathbb{E}[\hat{z}_i^j z_i^2 \hat{z}_i^k] = \mathbb{E}[\hat{z}_i^j z_i]\mathbb{E}[\hat{z}_i^k z_i] \tag{14}$$

with the fact that $z_i^2 = 1$. In Eq. (14), $\hat{\mathbb{E}}[\hat{z}_i^j \hat{z}_i^k] = \frac{1}{|x_u|}\sum_{x_u} \hat{z}_i^j \hat{z}_i^k$ is observable, which can be derived from the noisy labels of the $j$-th and $k$-th LFs, while $\mathbb{E}[\hat{z}_i^j z_i]$ and $\mathbb{E}[\hat{z}_i^k z_i]$ remain to be solved due to true label $z_i$ is unavailable. Next, we introduce a third labeling result from the $l$-th LF as $\hat{z}_i^l$, such that $\hat{\mathbb{E}}[\hat{z}_i^j \hat{z}_i^l]$ and $\hat{\mathbb{E}}[\hat{z}_i^k \hat{z}_i^l]$ are observable. Then, $|\hat{\mathbb{E}}[\hat{z}_i^j z_i]|$, $|\hat{\mathbb{E}}[\hat{z}_i^k z_i]|$, $|\hat{\mathbb{E}}[\hat{z}_i^l z_i]|$ can be solved by a triplet method as follows:

$$\begin{aligned}
|\hat{\mathbb{E}}[\hat{z}_i^j z_i]| &= \sqrt{|\hat{\mathbb{E}}[\hat{z}_i^j \hat{z}_i^k] \cdot \hat{\mathbb{E}}[\hat{z}_i^j \hat{z}_i^l] / \hat{\mathbb{E}}[\hat{z}_i^k \hat{z}_i^l]|}, \\
|\hat{\mathbb{E}}[\hat{z}_i^k z_i]| &= \sqrt{|\hat{\mathbb{E}}[\hat{z}_i^j \hat{z}_i^k] \cdot \hat{\mathbb{E}}[\hat{z}_i^k \hat{z}_i^l] / \hat{\mathbb{E}}[\hat{z}_i^j \hat{z}_i^l]|}, \\
|\hat{\mathbb{E}}[\hat{z}_i^l z_i]| &= \sqrt{|\hat{\mathbb{E}}[\hat{z}_i^j \hat{z}_i^l] \cdot \hat{\mathbb{E}}[\hat{z}_i^k \hat{z}_i^l] / \hat{\mathbb{E}}[\hat{z}_i^j \hat{z}_i^k]|}.
\end{aligned} \tag{15}$$

We can obtain the estimated accuracy of each LF by resolving the sign of $\mathbb{E}[\hat{z}_i^k z_i]$ [21]. Let $\hat{a}_i^k := \hat{P}(\hat{z}_i^k = z_i | \hat{z}_i^k \neq 0)$ be the estimated accuracy of the $k$-th LF on the $i$-th category. Therefore, the regularizer of $R(\theta, \hat{\mathbf{y}}_u)$ can be formulated as

$$R(\theta, \hat{\mathbf{y}}_u) = \sum_{i=1}^{C}\sum_{k}^{K} \hat{a}_i^k \log P_\theta(\hat{z}_i^k = z_i | \hat{z}_i^k \neq 0) + (1 - \hat{a}_i^k)\log(1 - P_\theta(\hat{z}_i^k = z_i | \hat{z}_i^k \neq 0)) \tag{16}$$

where $P_\theta(\hat{z}_i^k = z_i | \hat{z}_i^k \neq 0)$ can be computed in closed form by marginalizing over all the other variables in the model in Eq. (8). Details of $P_\theta$ can be referred to **Appendix** A.3.

### 3.6 End Model

In Step 3, probabilistic labels are used to train an end model under any network architecture. We utilize noise-aware empirical risk expectation as the objective function to take annotation errors into account. Accordingly, the final objective function is as follows:

$$\mathcal{L}(x_l, y_l, x_u, \pi) = \underbrace{\sum_{x_l} H(y_l, p(y|x_l))}_{\text{labeled samples}} + \underbrace{\sum_{x_u} \mathbb{E}_{y \sim \pi} H(y, p(y|x_u))}_{\text{unlabeled samples with probabilistic label}} \tag{17}$$

where $p(y|x_l)$ and $p(y|x_u)$ are the predicted distributions of $x_l$ and $x_u$, $\pi$ is the distribution produced by the label model in Sec. 3.4.

## 4 Experiments

### 4.1 Implementation Details

In the training phase, we follow the settings of previous works [4, 35, 36], augment data in weak (a standard flip-and-shift strategy) and strong forms (RandAugment [49] followed by Cutout [50] operation), and utilize a Wide ResNet as the end model for a fair comparison. In our framework, the batch size for labeled data and unlabeled data is set to 64 and 448, respectively. Besides, we use the same hyperparameters ($K = 50$, $\rho = 0.2$, $\epsilon = 0.95$) for all datasets. We compare DP-SSL with major existing methods on CIFAR-10 [53], CIFAR-100 [53], SVHN [54] and STL-10 [55]. We also analyze the effect of annotation and conduct ablation study in Sec. 4.4 and Sec. 4.5 respectively. All experiments are implemented in Pytorch v1.7 and conducted on 16 NVIDIA RTX3090s.

### 4.2 Datasets

**CIFAR-10 and CIFAR-100** [53] contain 50,000 training examples and 10,000 validation examples. All images are of 32x32 pixel size and fall in 10 or 100 classes, respectively.

**SVHN** [54] is a digital image dataset that consists of 73,257, 26,032 and 531,131 samples in the train, test, and extra folders. It has the same image resolution and category number as CIFAR-10.

Table 1: Results of error rate on CIFAR-10, CIFAR-100 and SVHN for different existing SSL methods (Π-Model [29], Pseudo-Labeling [7], Mean Teacher [32], MixMatch [31], UDA [34], ReMixMatch [35], FixMatch [4] and USADTM [36]) and our DP-SSL method.

| Method | CIFAR-10 | | | CIFAR-100 | | | SVHN | | |
|---|---|---|---|---|---|---|---|---|---|
| | 40 labels | 250 labels | 4000 labels | 400 labels | 2500 labels | 10000 labels | 40 labels | 250 labels | 1000 labels |
| Π -Model | - | 54.26±3.97 | 14.01±0.38 | - | 57.25±0.48 | 37.88±0.11 | - | 18.96±1.92 | 7.54±0.36 |
| Pseudo-Labeling | - | 49.78±0.43 | 16.09±0.28 | - | 57.38±0.46 | 36.21±0.19 | - | 20.21±1.09 | 9.94±0.61 |
| Mean Teacher | - | 32.32±2.30 | 9.19±0.19 | - | 53.91±0.57 | 35.83±0.24 | - | 3.57±0.11 | 3.42±0.07 |
| MixMatch | 47.54±11.50 | 11.05±0.86 | 6.42±0.10 | 67.61±1.32 | 39.94±0.37 | 28.31±0.33 | 42.55±14.53 | 3.98±0.23 | 3.50±0.28 |
| UDA | 29.05±5.93 | 8.82±1.08 | 4.88±0.18 | 59.28±0.88 | 33.13±0.22 | 24.50±0.25 | 52.63±20.51 | 5.69±2.76 | 2.46±0.24 |
| ReMixMatch | 19.10±9.64 | 5.44±0.05 | 4.72±0.13 | 44.28±2.06 | **27.43**±0.31 | 23.03±0.56 | 3.34±0.20 | 2.92±0.48 | 2.65±0.08 |
| USADTM | 9.54±1.04 | 4.80±0.32 | 4.40±0.15 | 43.36±1.89 | 28.11±0.21 | **21.35**±0.17 | 3.01±1.97 | **2.11**±0.65 | **1.96**±0.05 |
| FixMatch (RA) | 13.81±3.37 | 5.07±0.65 | 4.26±0.05 | 48.85±1.75 | 28.29±0.11 | 22.60±0.12 | 3.96±2.17 | 2.48±0.38 | 2.28±0.11 |
| FixMatch (CTA) | 11.39±3.35 | 5.07±0.33 | 4.31±0.15 | 49.95±3.01 | 28.64±0.24 | 23.18±0.11 | 7.65±7.65 | 2.64±0.64 | 2.36±0.19 |
| DP-SSL (ours) | **6.54**±0.98 | **4.78**±0.26 | **4.23**±0.20 | **43.17**±1.29 | 28.00±0.79 | 22.24±0.31 | **2.98**±0.86 | 2.16±0.36 | 1.99±0.18 |
| Fully Supervised | | 2.74 | | | 16.84 | | | 1.48 | |

Table 2: Results of error rate on STL-10.

| | STL-10 | | | | | | |
|---|---|---|---|---|---|---|---|
| Method | 1000 labels | Method | 1000 labels | Method | 40 labels | 250 labels | 1000 labels |
| Π -Model | 26.23±0.82 | UDA | 7.66±0.56 | USADTM | 9.63±1.35 | 6.85±1.09 | **4.01**±0.59 |
| Pseudo-Labeling | 27.99±0.80 | ReMixMatch | 5.23±0.45 | DP-SSL (ours) | **9.32**±0.91 | **6.83**±0.71 | 4.97±0.42 |
| Mean Teacher | 21.43±2.39 | FixMatch (RA) | 7.98±1.50 | Fully Supervised | | 1.48 | |
| MixMatch | 10.41±0.61 | FixMatch (CTA) | 5.17±0.63 | | | | |

**STL-10** [55] is a dataset for evaluating unsupervised and semi-supervised learning. It consists of 5000 labeled images and 8000 validation samples of 96x96 size from 10 classes. Besides, there are 100,000 unlabeled images available, including odd samples.

### 4.3 Comparison with Existing SSL Methods

For a fair comparison, we conduct experiments with the codebase of FixMatch and cite the results on CIFAR-10, CIFAR-100, SVHN and STL-10 from [4, 36]. We utilize the same network architecture (a Wide ResNet-28-2 for CIFAR-10 and SVHN, WRN-28-8 for CIFAR-100, and WRN-37-2 for STL-10) and training protocol of FixMatch, such as optimizer and learning rate schedule. Unlabeled data are generated by the scripts in FixMatch. Results of DP-SSL and existing methods in Tab. 1 and Tab. 2 are presented with the mean and standard deviation (STD) of accuracy on 5 pre-defined folds.

As shown in Tab. 1, our method achieves the best performance in most cases, especially when there are only 4 labeled samples per class. Specifically, our method achieves a 93.46% accuracy on CIFAR-10 with 4 labeled samples per category, which is 3.3% higher than that of USADTM — the state-of-the-art method. Again on STL-10, our method surpasses USADTM and achieves the best performance when there are 4 and 25 labeled samples per class.

On CIFAR-100, our method performs the best for 400 labels case and the 2nd for 2500 and 10,000 labels cases. We also notice that DP-SSL has relatively large STDs for 2500 and 10,000 labels cases, which is due to the coarse accuracy estimation. In fact, even if triplet mean is adopted in estimation, the triplet selection in Eq. (15) still impacts accuracy estimation and regularizer a lot, especially when $\mathbb{E}[\hat{z}_i^k z_i]$ is close to 0 or sign recovery of $\mathbb{E}[\hat{z}_i^k z_i]$ is wrong. Actually, there are some advanced approaches to unsupervised accuracy estimation [56–58] that can replace the naive triplet mean estimation. Ideally, if we can obtain the exact accuracy of each class $\hat{b}_i^k := \hat{P}(\hat{z}_i^k = z_i | \hat{z}_i^k = 1)$ and regularize it as $R(\theta, \hat{\mathbf{y}}_u) = \sum_{i=1}^C \sum_k^K \hat{b}_i^k \log P_\theta(\hat{z}_i^k = z_i | \hat{z}_i^k = 1) + (1 - \hat{b}_i^k) \log(1 - P_\theta(\hat{z}_i^k = z_i | \hat{z}_i^k = 1))$, we will get an end model with $(27.92 \pm \mathbf{0.23})\%$ error rate for 2500 labeled samples.

Comparing with USADTM, our method does not perform well enough when more labeled data available. For USADTM, apart from the proxy label generator, unsupervised representation learning contributes a lot for its performance. As shown in the ablation study of [36], USADTM without unsupervised representation learning achieves around 5.73% and 4.99% error rate for 250 and 4000 labeled samples in CIFAR-10, while our method DP-SSL obtains 4.78% and 4.23% error rate.

Table 3: The macro Precision/Recall/F1 Score/Coverage of the annotated labels on CIFAR-10, CIFAR-100, and SVHN for our method and two typical existing label models.

| Method | Metrics | CIFAR-10 | | | CIFAR-100 | | | SVHN | | |
|---|---|---|---|---|---|---|---|---|---|---|
| | | 40 labels | 250 labels | 4000 labels | 400 labels | 2500 labels | 10000 labels | 40 labels | 250 labels | 1000 labels |
| Majority Vote | F1 Score | 85.96 | 94.23 | 95.77 | 49.97 | 69.81 | 76.03 | 90.86 | 95.38 | 96.14 |
| FlyingSquid[21] | F1 Score | 90.25 | 94.99 | 95.85 | 48.90 | 69.73 | 74.12 | 93.92 | 97.24 | 97.70 |
| DP-SSL (ours) | Precision | 93.47 | 95.30 | 95.89 | 55.62 | 71.91 | 75.12 | 95.20 | 97.65 | 97.79 |
| | Recall | 93.82 | 95.33 | 95.91 | 56.86 | 72.01 | 78.35 | 96.78 | 97.64 | 97.94 |
| | F1 Score | **93.61** | **95.19** | **95.90** | **54.42** | **71.89** | **76.36** | **95.95** | **97.59** | **97.81** |
| | Coverage | 99.35 | 99.79 | 99.91 | 99.33 | 99.87 | 99.94 | 99.15 | 99.67 | 99.93 |

## 4.4 Analysis

**Annotation performance**. Intuitively, the holistic performance of the end model in our method highly depends on the quality of annotation results. Thus, we present the macro precision/recall/F1 score and coverage of the annotated labels of our method on CIFAR-10, CIFAR-100, and SVHN in Tab. 3. We can see that our method achieves over 99% coverage, which means that it produces probabilistic labels for almost all unlabeled data. Comparing to the results in [36], the label model with 40 labeled samples outperforms the proxy label generator, FixMatch and USADTM get 88.51% and 89.48% accuracy, respectively. Furthermore, our method achieves 97.36% accuracy for unlabeled data with the top-500 highest probabilities in each category. Meanwhile, we also present results of Majority Voting and FlyingSquid [21] in Tab. 3 based on the noisy labels from Step 1 of our method for comparison. Majority Voting gets bad performance because the number of LFs triggered for different categories is not equal. For FlyingSquid, we implement it with $C$ one-versus-all models to support multi-class tasks, and the large $C$ in CIFAR-100 results in the worst performance.

**Barely supervised learning**. We conduct experiments to test the performance (accuracy and STD) of our method on CIFAR-10 for some extreme cases (10, 20 and 30 labeled samples) to verify the effectiveness of our method. Here, we select the labeled data through the scripts of FixMatch with 5 different random seeds. As claimed in FixMatch, it reaches between 48.58% and 85.32% test accuracy with a median of 64.28% for 10 labeled samples, while our method obtains accuracy from 61.32% to 83.7%. As for 20 and 30 labeled samples, our method gets $(85.29 \pm 3.14)\%$ and $(89.81 \pm 1.59)\%$ accuracy respectively, which have much smaller STDs than that reported in [37].

## 4.5 Ablation Study

In DP-SSL, LFs and the label model are the core components to assign probabilistic labels for training the end model. Here, we check the effects of the following factors in the process of producing probabilistic labels by taking CIFAR-10 as the example. For ease of exposition, only the accuracy of predicted labels is presented in Tab. 4.

Table 4: Annotation performance for different configurations on CIFAR-10 with 40 and 250 labels. $K$ and $\rho$ are set to 50 and 0.2 by default.

| Experiments | 40 labels | 250 labels |
|---|---|---|
| Exp1: *w.o.* MCL | 92.46 | 95.02 |
| Exp2: MCL *w.o.* FT | 91.61 | 94.98 |
| Exp3: MCL *w.* FT | **93.82** | **95.33** |
| Exp4: *w.o.* Regularizer | 93.19 | 94.94 |
| Exp5: Regularizer | **93.82** | **95.33** |

**MCL**. Feature transformation (FT) described in Eq. (3) can be regarded as a weighted spatial pooling for extracted features. It is proposed to boost the diversity of generated LFs. We conduct comparative experiments for three configurations: 1) *Exp1*: *w.o.* MCL, 2) *Exp2*: MCL *w.o.* FT, 3) *Exp3*: MCL *w.* FT. The results are presented in Tab. 4. It is interesting to see that *Exp1* is better than *Exp2* but worse than *Exp3*. In fact, *Exp1* is a simple ensemble model with a shared backbone, where each LF is trained independently and predicts the labels within $C$ categories. In *Exp2*, we observe that some classifiers have never been optimized in the training phase and thus have an empty specialized set when only a few labeled samples per class are available. Moreover, the specialized sets of many LFs are duplicate, which incurs a negative impact on the performance. However, MCL with FT addresses the drawbacks and helps our method obtain versatile LFs.

**Hyperparameters**. $K$ and $\rho$ are the number of LFs and the ratio of specialists in Eq. (4). In our ablation study, we focus on the variance of performance for different $K$ and $\rho$ with 40 labeled samples

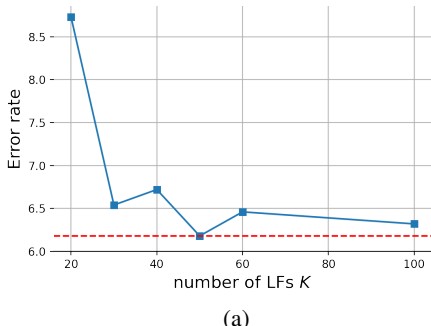
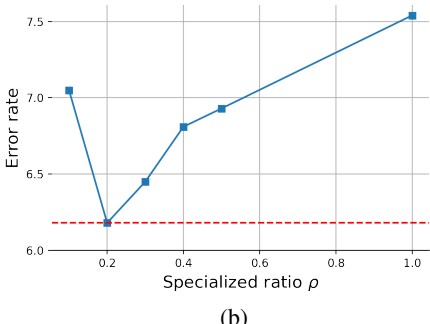

|          (a)          |          (b)          |

Figure 3: Plots of ablation study on CIFAR-10 with 40 labels. (a) Varying the number of LFs K. (b) Varying the specialized ratio $\rho$. Here, the red dashed line indicates the error rate of DP-SSL with default hyperparameters.

on CIFAR-10. In Fig. 3a, $K$=50 performs the best when 40 labeled samples are available. On the other hand, performance reaches the best when $\rho$=0.2 in Fig. 3b. We present more results of $\rho$ and $K$ in Appendix A.5.

**Regularizer**. The regularizer is proposed to impose a global guidance and improve the robustness of the label model. As shown in Tab. 4, the regularizer does boost the accuracy, especially when facing less labeled samples. Besides, as mentioned in Sec. 4.3, the high-quality guidance of the regularizer also reduces the label model's performance variance, thus improves its robustness.

## 5    Conclusion

In this paper, we explore the data programming idea to boost SSL when only a small number of labeled samples available by providing more accurate labels for unlabeled data. To this end, we propose a new SSL method DP-SSL that employs an innovative DP mechanism to automatically generate labeling functions. To make the labeling functions diverse and specialized, a multiple choice learning based approach is developed. Furthermore, we design an effective label model by incorporating a novel potential and a regularizer with estimated accuracy. With this model, probabilistic labels are inferred by resolving the conflict and overlap among noisy labels from the labeling functions. Finally, an end model is trained under the supervision of the probabilistic labels. Extensive experiments show that DP-SSL can produce high-quality probabilistic labels, and outperforms the existing methods to achieve a new SOTA, especially when only a small number of labeled samples are available.

## 6    Limitations of This Work

In this work, we use coarse accuracy estimation as the statistic information to guide the label model for simplicity. As described in Sec. 3.5, we estimate the accuracy $P_\theta(\hat{z}_i^k = z_i | \hat{z}_i^k \neq 0)$, rather than class-wise accuracy $P_\theta(\hat{z}_i^k = z_i | \hat{z}_i^k = 1)$. Besides, we do not consider the dependency between LFs and directly assume they are independent.

## Acknowledgments and Disclosure of Funding

This work was supported by Alibaba Group through Alibaba Innovative Research Program. Shuigeng Zhou was also partially supported by Science and Technology Commission of Shanghai Municipality Project (No. 19511120700), and Shanghai Artificial Intelligence Innovation and Development Projects funded by Shanghai Municipal Commission of Economy and Informatization.

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
