Table 5: Glossary of variables and symbols used in this paper.

| Symbol | Used for |
|--------|----------|
| $x_l, y_l$ | Labeled samples and ground-truth labels |
| $x_u$ | Unlabeled samples |
| $x_{\cdot}^{w}$ | Weakly augmented (labeled / unlabeled) samples |
| $x_{\cdot}^{s}$ | Strongly augmented (labeled / unlabeled) samples |
| $\mathcal{Y}$ | Label space. *e.g.*, in CIFAR-10, $\mathcal{Y} = \{1, 2, \cdots, 10\}$ |
| $C$ | Number of categories. $C = |\mathcal{Y}|$ |
| $f$ | $\mathbb{R}^{HW \times D}$, feature map before global average pooling of backbone |
| $f(j)$ | $\mathbb{R}^{D}$, feature vector at spatial position $j$ of $f$ |
| $dis(A, B)$ | Distance between A and B |
| $c_k$ | $\mathbb{R}^{D}$, learnable clustering center of the $k$-th LF |
| $f_k$ | $\mathbb{R}^{D}$, input feature of classifier in the $k$-th LF |
| $\mathcal{F}_k$ | The classifier head in the $k$-th LF |
| $p_k(y|x)$ | $\mathbb{R}^{C}$, output probability of the $k$-th LF |
| $\tau_k$ | Specialized category set of the $k$-th LF |
| $\bar{p}_k(y|x)$ | $\mathbb{R}^{|\tau_k|+1}$, output probability over specialized categories and "abstention" option |
| $\hat{y}^k$ | $= \arg max(\bar{p}_k(y|x))$, predicted label of the $k$-th LF |
| $\hat{\mathbf{y}}$ | $= (\hat{y}^1, \cdots, \hat{y}^K)^{\mathsf{T}} \in \mathbb{R}^{K}$, vectorized predicted laebls of K LFs |
| $\theta$ | $\theta \in \mathbb{R}^{K \times |\mathcal{Y}|}$ is the parameters of label model |
| $e_{ky}$ | $e_{ky} := exp(\theta_{ky})$, exponent of parameters $\theta_{ky}$ |
| $\phi(y, \hat{y}^k)$ | Potential value with the target label $y$ and predicted label $\hat{y}^k$ |
| $P(y, \hat{\mathbf{y}})$ | Joint distribution between target label $y$ and predicted label $\hat{\mathbf{y}}$ in the label model |
| $Z$ | Normalizer item of $P(y, \hat{\mathbf{y}})$. $Z = \sum_{y \in \mathcal{Y}} \sum_{\hat{\mathbf{y}} \in \tau} P(y, \hat{\mathbf{y}})$ |
| $R(\theta, \hat{\mathbf{y}}_u)$ | Regularizer of unlabeled data in the Label Model |
| $z_i$ | $z_i = \begin{cases} +1 & y_u = i \\ -1 & y_u \neq i \end{cases}$, latent variable of ground-truth label in the $i$-th one-versus-all task |
| $\hat{z}_i^k$ | $\hat{z}_i^k = \begin{cases} +1 & \hat{y}^k = i \\ 0 & \hat{y}^k = 0 \\ -1 & otherwise \end{cases}$, latent variable of $\hat{y}^k$ in the $i$-th one-versus-all task |
| $\mathbb{E}[\hat{z}_i^k z_i]$ | Expectation of $\hat{z}_i^k z_i$ |
| $\hat{\mathbb{E}}[\hat{z}_i^k z_i]$ | Estimated expectation of $\hat{z}_i^k z_i$ over all unlabeled data without ground-truch $z_i$. |
| $\hat{P}(\cdot)$ | Probability estimated with the observable data. |
| $\hat{a}_i^k$ | Precision of the $k$-th LF in the $i$-th one-versus-all classification |

# A   Appendix

## A.1   Glossary

The glossary is given in Table 5.

## A.2   Label Model

Label Model in Snorkel [52] is also called as "Generative Model", which models and integrates the noisy labels provided by $K$ LFs. In this paper, we suppose that the $K$ LFs are independent. Assuming that $\hat{\mathbf{y}} = (\hat{y}_1, \cdots, \hat{y}_K)^{\mathsf{T}} \in \mathbb{R}^{K}$ is the vectorized form of the predicted labels from $K$ LFs, and $\hat{y}_k$ is the predicted label of the $k$-th model. For clarity, we denote $\emptyset$ as the "abstention" option in the LFs. Then, following the definition of Snorkel, the label model in Snorkel can be represented as:

$$P(y, \hat{\mathbf{y}}) = \frac{1}{Z} \prod_{k=1}^{K} \phi(y, \hat{y}^k), \tag{18}$$

with the potential function $\phi(y, \hat{y}^k)$:

$$\phi(y, \hat{y}^k) = \begin{cases} exp(\theta_{k1} + \theta_{k2}), & if\ \hat{y}^k = y \\ exp(\theta_{k1}), & if\ \hat{y}^k \neq y, \hat{y} \neq \emptyset \\ 1. & if\ \hat{y}^k = \emptyset \end{cases} \tag{19}$$

where $\theta \in \mathbb{R}^{2K}$. It can be observed that the potential function in Eq. (19) provides the same values for all target categories $y$. However, each LF in our method specializes in multiple categories with different performance. Thus, we extend the parameters $\theta$ to $K \times C$ to support multi-class classification. In Eq. (19), we set $exp(\theta_{k1} + \theta_{k2})$ to guarantee that the potential with $\hat{y}^k = y$ is larger than that with $\hat{y}^k \neq y$. Similarly, we set $1 + exp(\theta_{ky})$ and $1/(1 + exp(\theta_{ky}))$. Then, we have the potential function:

$$\phi(y, \hat{y}^k) = \begin{cases} 1 + exp(\theta_{yk}), & if\ \hat{y} \in \tau_k,\ \hat{y}^k = y \\ 1/(1 + exp(\theta_{yk})), & if\ \hat{y} \in \tau_k,\ \hat{y}^k \neq y \\ 1. & otherwise \end{cases} \tag{20}$$

However, the conclusion from [23, 25] tells us that MCL tends to be overconfident for the samples whose ground-truth labels are out of the specialized set. In other words, when one LF is fed with a sample whose ground-truth label is out of the specialized set, it may still produce a label in the specialized set with high confidence. Although the cases of overconfidence decrease a lot due to the introduction of the "abstention" option in Step 1, these cases still exist in our framework. Therefore, we introduce the item $exp(\theta_{yk})$ to represent the relationship between the predicted label $\hat{y}^k$ and the target label $y$, even when the target labels conflict with the predicted labels. Based on these considerations, we define the four-part potential function in Eq. (9.

## A.3 Regularizer

We give the complementary formulation of $P_\theta(\hat{z}_i^k = z_i | \hat{z}_i^k \neq 0)$. Set $\Phi^k(y, \hat{\mathcal{Y}}) := \sum_{\hat{y}^k \in \hat{y}} \phi(y, \hat{y}^k)$ for ease of exposition. We write $P_\theta(\hat{z}_i^k = z_i | \hat{z}_i^k \neq 0)$ as follows:

$$
\begin{aligned}
&P_\theta(\hat{z}_i^k = z_i | \hat{z}_i^k \neq 0) \\
&= \frac{P_\theta(\hat{z}_i^k = z_i = 1) + P_\theta(\hat{z}_i^k = z_i = -1)}{P_\theta(\hat{z}_i^k \neq 0)} \\
&= \frac{P_\theta(y = i, \hat{y}^k = y, \hat{y}^k \neq 0) + P_\theta(y \neq i, \hat{y}^k \neq i, \hat{y}^k \neq 0)}{P_\theta(\hat{y}^k \neq 0)} \\
&= \frac{\phi(i, i) \prod_{k' \neq k} \Phi^{k'}(i, \{0\} \cup \tau_{k'}) + \sum_{y \neq i} (\Phi^k(y, \tau_k - \{i\}) \prod_{k' \neq k} \Phi^{k'}(y, \{0\} \cup \tau_{k'}))}{\sum_{y \in \mathcal{Y}} \Phi^k(y, \tau_k) \prod_{k' \neq k} \Phi^{k'}(y, \{0\} \cup \tau_{k'})}.
\end{aligned}
\tag{21}
$$

As mentioned in our paper, the class-wise accuracy $P_\theta(\hat{z}_i^k = z_i | \hat{z}_i^k = 1)$ without negative samples can be written as:

$$P_\theta(\hat{z}_i^k = z_i | \hat{z}_i^k = 1) == \frac{\phi(i, i) \prod_{k' \neq k} \Phi^{k'}(i, \{0\} \cup \tau_{k'})}{\sum_{y \in \mathcal{Y}} \phi(y, i) \prod_{k' \neq k} \Phi^{k'}(y, \{0\} \cup \tau_{k'})}. \tag{22}$$

However, we can estimate $\hat{P}(\hat{z}_i^k = z_i | \hat{z}_i^k \neq 0)$ by Eq. (15) directly, but it is more difficult to estimate the class-wise accuracy $\hat{P}(\hat{z}_i^k = z_i | \hat{z}_i^k = 1)$.

## A.4 Model Analysis

We give an example with 40 labeled samples in Fig. 4 to illustrate why Majority Vote falls with $K = 50$ and $\rho = 0.2$. In the extreme case, category "automobile" and category "ship" are only specialized by 4 LFs, while 15 LFs specialize in "dog". If an image with category "automobile" triggers all (4) true specialized LFs but triggers 40% (5) specialized LFs with class "dog", Majority Vote would misclassify it into "dog". With our label model, we can achieve 95.22% annotation accuracy in this case.

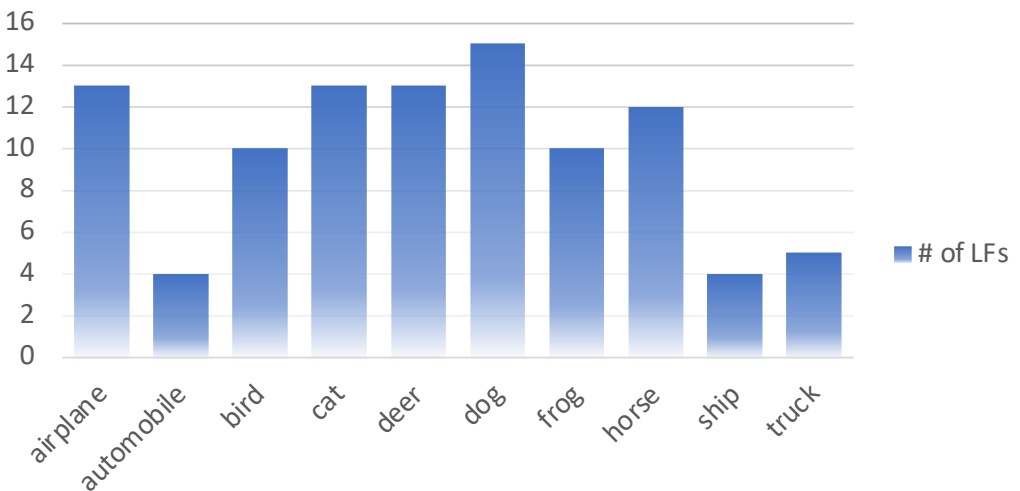

Figure 4: The number of specialized LFs for each category.

Table 6: Annotation performance for different $\rho$ and $K$ on CIFAR-10 with 40 labels

|    | 0.1 | 0.2 | 0.3 | 0.4 | 0.5 | 1.0 |
|----|------|------|------|------|------|------|
| 10 | 83.25 | 86.42 | 88.99 | 90.62 | 91.28 | 92.39 |
| 20 | 89.16 | 91.27 | 91.83 | 92.31 | 92.52 | 92.48 |
| 30 | 91.39 | 93.46 | 93.54 | 93.16 | 92.91 | 92.41 |
| 40 | 92.53 | 93.28 | 93.45 | 93.24 | 92.80 | 92.47 |
| 50 | 92.95 | 93.82 | 93.55 | 93.19 | 93.07 | 92.46 |
| 60 | 93.01 | 93.54 | 93.11 | 92.97 | 92.63 | 92.43 |

## A.5 Hyperparameter $\rho$ and $K$

We also present the complete experimental results with different $\rho$ and $K$ on CIFAR-10 with 40 labels in Tab. 6.