# OpenReview forum: "DP-SSL: Towards Robust Semi-supervised Learning with A Few Labeled Samples"
_NeurIPS.cc/2021/Conference — NeurIPS 2021 Poster_

### Official Review · Reviewer_cBn7 · 2021-07-14

**Rating:** 6
**Confidence:** 3

**Summary:**

This paper focuses on the extreme case of semi-supervised learning where very few labeled samples are available (e.g. 40 for CIFAR-10). In this extreme case, pseudo-labels for self-training are noisy and unreliable. Therefore, the authors proposed to first learn a set of labeling functions (LFs) that only focus on classifying a subset of classes, and then aggregate the predictions from the LFs for generating more reliable pseudo-labels.

In the experiments, they demonstrate superior performance in the case of very few labeled data on CIFAR-10, CIFAR-100, SVHN, and STL-10. In the cases of more labeled data, the proposed method achieves comparable performance with the previous SoTA.

**Limitations And Societal Impact:**

- In Tab.1 the proposed method does not perform much better on CIFAR-100. Does that mean the proposed method does not generalized well to larger datasets or datasets with more classes?

- In SSL, the trained model tends to bias toward the selected labeled data. In the case of extremely few labeled data, the bias introduced by these few labeled data is magnified.

**Main Review:**

- Despite its great performance in the setting of extremely few labeled data, the method is not clearly explained:

  - Eq. 1 is introduced in Sec. 3.2 but has never been used in the rest of Sec. 3.
  - It is unclear which arguments or parameters the proposed loss functions are optimized with respect to.
  - $\theta$ at L188 is undefined
  - In the first term of Eq.9, P(y, $\tilde{y}_l$) is a joint distribution of y and $\tilde{y}_l$, which is expected to be CxK. How is loss computed with a target $y_l$ of dimension C?
  - In the second term of Eq. 9, the purpose for minimizing the negative log marginal likelihood is not explained.
  - How to train these LFs? Is each step trained sequentially with models trained in previous steps fixed? Or are they all trained jointly?
  - Method like FixMatch uses a confidence score to filter out unconfident pseudo-labels. Does the proposed method use ALL pseudo-labels without any confidence thresholding?

- How to optimize Eq.2? According to the reviewer's understanding, gradient descent is not the best option for constrained optimization. However, it looks like Eq.2 is optimized jointly with other losses in Eq.5. Moreover, $u^k$ are discrete variables, which are not optimizable by gradient descent. Again, without knowing which arguments or parameters the loss is optimized with respect to, it's hard to understand how the proposed method can automatically learn a good set of LFs by optimizing Eq.2.

- More analyses are expected for the learned LFs. For example, the authors claim that "each LF is a specialist for some classes, so it can get high accuracy for samples in these classes" at L163. Perhaps it's a good idea to show what are the learned LFs and how's their performance.

**Time Spent Reviewing:**

4

---

> ### Author Response · Authors · 2021-08-10
> **Response to Reviewer cBn7**
>
> Q1. **Eq.1 has never been used in the rest of Sec. 3.**
>
> Response: Actually, feature transformation only works between the backbone and classifiers. It transforms the feature for multiple classifiers. Then, the $k$-th classifier further takes the transformed feature $\tilde{f}_k$ and outputs the probability $\tilde{p}^k$. In the rest of Sec.3, we focus on the output probability $\tilde{p}^k$, which is dependent on $\tilde{f}_k$ in Eq. (1).  So Eq. 1 is actually used in the rest of Sec. 3.
>
> supp: For notation $\tilde{p}^k$, another subscript $\in\lbrace l,u\rbrace$ and the superscript $\in \lbrace w,s\rbrace$ are omitted for clear presentation.
>
> Q2. **It is unclear which arguments or parameters the proposed loss functions are optimized with respect to.**
>
> Response: Our model is trained step-by-step. Therefore, the loss function in Eq. (5) affects the shared backbone and multiple classifiers. The objective function in Eq. (9) is used to optimize the parameter $\theta$ in the label model. As for the third loss function, it is used to train the parameters in the end model.
>
> Q3. **$\theta$ at L.188 is undefined.**
>
> Response: $\theta$ is the parameters of the label model, which is explained in L.188. We have also given $e_{ky}:=exp(\theta_{ky})$ in L.189, where $\theta_{ky}$ is the parameter at the position $(k,y)$ of $\theta$.
>
> Q4. **$P(y, \tilde{\mathbf{y}}_l)$​​​ in Eq.(9) is expected to be CxK. How is the loss computed with a target $\tilde{\mathbf{y}}_l$​​​ of dimension C?**
>
> Response: We train the label model after obtaining the converged LFs. Thus, we get a deterministic prediction $\tilde{\mathbf{y}}_l$ from Step 1. With a deterministic prediction $\tilde{\mathbf{y}}_l$, $P(y,\tilde{\mathbf{y}}_l)$ in Eq.(9) is expected to be Cx1, the dimension of K is aggregated by the product operator in Eq.(6). Then,  $P(y,\tilde{\mathbf{y}}_l) \in \mathbb{R}^{C\times 1}$ is applied into the cross-entropy $H(y_l, P(y,\tilde{\mathbf{y}}_l))$.
>
> Q5. **In the second term of Eq. (9), the purpose for minimizing the negative log marginal likelihood is not explained.**
>
> Response: From the definition of joint distribution in Eq. (6), we can obtain $\sum_{y \in \mathcal{Y}}\sum_{\forall \tilde{\mathbf{y}}}P(y,\tilde{\mathbf{y}})=1$. Therefore, minizing the negative log marginal likelihood can help maximize $\sum_{y\in \mathcal{Y}} P(y,\tilde{\mathbf{y}})$ with a deterministic term $\tilde{y}_u$, which can be regarded as maximizing the likelihood on the observed data.
>
> Q6. **How to train these LFs? Is each step trained sequentially with models trained in previous steps fixed? Or are they all trained jointly?**
>
> Response: First, all LFs in our method are trained jointly. Concretely, as claimed in L.175-178, we first train LFs jointly with $\mathcal{L}_1(x_l^w,y_l)$ to obtain the specialized set automatically, then use the other two losses $\mathcal{L}_2$ and $\mathcal{L}_u$ to optimize the models in a semi-supervised manner.
>
> Second, each step is trained sequentially with the fixed models trained in the previous step.
>
> Q7. **Does the proposed method use all pseudo-labels without any confidence thresholding?**
>
> Response: No, we do use confidence thresholding in Eq. (4) in step 1.
> In Step 1, pseudo-labels are filtered through a pre-defined confidence threshold $\epsilon$, as claimed in Eq. (4) and L.174.
> In Step 2, we take the prediction $\tilde{\mathbf{y}} \in \mathbb{R}^K$ as the input. Since a large number of LFs have been trained in the previous step, the label model can produce probabilistic labels for the next step according to conflict or agreement between different dimensions of $\tilde{\mathbf{y}}$.
> In Step 3, we utilize the noisy-aware empirical risk expectation on unlabeled samples to train the end model.
>
> Q8. **How to optimize Eq.(2), more explanations, especially for $u^k$​​? Without knowing which arguments or parameters the loss is optimized with respect to, it's hard to understand how the proposed method can automatically learn a good set of Lfs by optimizing Eq.(2).**
>
> Response: Generally, only top-($\rho*K$) small losses among $K$ LFs are utilized to optimize the model for each sample.
>
> Take $K=4$​​, $\rho=0.5$​​ for example, given two labeled samples $x_1$​​ and $x_2$​​, the corresponding cross entropy of 4 LFs are [0.1, 0.4, 3, 5] and [0.4, 0.1, 0.2, 2], respectively. Thus, the indicator variable $u$​​ is derived as [1, 1, 0, 0] and [0, 1, 1, 0] via top-$\rho\times K$​ small operators. Afterwards, LFs with top-$(\rho*K)$​ small losses are optimized.  Concretely, $LF_1$​ and $LF_2$​ are optimized with corresponding cross-entropy of $x_1$​, $LF_2$​ and $LF_3$​ are optimized with corresponding cross-entropy of $x_2$​​​​. After some iterations, each LF will tend to predict the similar samples with a high accuracy.
>
> Q9. **More analyses for the learned LFs. For example, the authors claim that "each LF is a specialist for some classes, so it can get high accuracy for samples in these classes" at L163. Perhaps it's a good idea to show what are the learned LFs and how's their performance.**
>
> Response: For better understanding, here we present the class-wise accuracy of some LFs of our model trained on CIFAR-10 with 40 labels in the following table. We can observe that each learned LF has a good accuracy of samples with the specialized classes.
>
> |            |   $LF_1$    |        $LF_2$         |      $LF_3$       |         $LF_4$         |
> | :--------: | :---------: | :-------------------: | :---------------: | :--------------------: |
> |  Airplane  |      0      |         0.970         |         0         |           0            |
> | Automobile |      0      |           0           |         0         |           0            |
> |    Bird    |      0      |           0           |         0         |         0.925          |
> |    Cat     |      0      |           0           |         0         |           0            |
> |    Deer    |      0      |           0           |         0         |           0            |
> |    Dog     |      0      |           0           |         0         |         0.941          |
> |    Frog    |      0      |           0           |       0.991       |           0            |
> |   House    |    0.977    |           0           |       0.981       |           0            |
> |    Ship    |      0      |           0           |         0         |           0            |
> |   Truck    |      0      |         0.972         |         0         |         0.977          |
> |  $\tau_k$​  | $\lbrace House\rbrace$ | $\lbrace Airplane, Truck\rbrace$​ | $ \lbrace Frog, House\rbrace$ | $ \lbrace Bird, Dog, Truck\rbrace$ |
>
> Q10. **In Tab.1, the proposed method does not perform much better on CIFAR-100. Does that mean the proposed method does not generalize well to larger datasets or datasets with more classes?**
>
> Response: Generally, it is more difficult to make decisions on datasets with more classes, and actually, almost all machine learning methods have worse performance on datasets with more classes. Besides, our method is implemented over FixMatch (RA), the most meaningful baseline should be FixMatch (RA). For this baseline, we reduce the error rate from $48.85$% to $43.17$%. As for USADTM, apart from the proxy label generator, there exists an unsupervised mutual information loss in USADTM, which can also bring improvement for FixMatch and our method.
>
> As most existing methods were tested on four widely-used datasets: CIFAR-10, CIFAR-100, SVHN, and STL-10, for a fair comparison, we also evaluated our method on these four datasets. However, we believe that our model is consistently capable of boosting the performance on large-scale datasets such as ImangeNet, and we will add empirical results on ImageNet to our revised version.
>
>
>
> Q11. **In SSL, the trained model tends to bias toward the selected labeled data. In the case of extremely few labeled data, the bias introduced by these few labeled data is magnified.**
>
> Response: Generally, we agree with you. However, we think the unsupervised data are helpful for counteracting the bias if the unlabeled are properly used and the SSL algorithm is well designed. In our paper, we have evaluated our method under 5 pre-defined folds for a fair comparison, and present the mean and standard deviation of the accuracy in Tab. 1 and Tab. 2. It can be seen that our method achieves higher accuracy with a much smaller standard deviation, which means our method is more robust than the existing methods.

---

> > ### Author Response · Authors · 2021-08-30
> > **Response to Reviewer cBn7**
> >
> > Dear Reviewer cBn7,
> >
> > We have properly addressed all your concerns and provided clarifications on all confusing concepts. Could you please kindly re-evaluate our paper based on the current situation? If you have any further questions, we are also very glad to discuss them.
> >
> > Thanks,
> >
> > Authors

---

> > > ### Comment · Reviewer_cBn7 · 2021-08-31
> > > **Score update**
> > >
> > > Thanks for the explanation. The reviewer's main concern is mostly about clarity. The authors have addressed most of the reviewer's concerns. Therefore, the reviewer decided to update the score from 4 to 6.

---

### Official Review · Reviewer_8Nkc · 2021-07-15

**Rating:** 6
**Confidence:** 3

**Summary:**

This paper tackles semi-supervised learning with a few labeled samples per class available during training. To solve this problem, the authors propose a data programming (DP) scheme to generate probabilistic labels for unlabeled data. Specifically, a multiple-choice learning (MCL) approach is proposed to generate a number of diverse and specialized labeling functions (LFs), which produce soft targets that indicate the class distributions of the unlabeled data. The proposed method is evaluated on four image classification datasets including CIFAR10, CIFAR100, SVHN, STL-10 in comparison to existing state-of-the-art methods.

**Limitations And Societal Impact:**

Although the paper includes a discussion of the limitations of the method, a broader discussion about the potential negative societal impacts is not given.

**Main Review:**

**Method**

The proposed method is interesting; however, it is unclear why it is better than existing methods like FixMatch. Several concerns are summarized in the following.

1. It is indicated that the label space are divided into multiple subsets and different classifiers (i.e. labeling functions (LFs)) are trained to predict only a subset of classes. However, it is unclear how the division is defined. How many LFs are required? How does the randomness in division of class space affects the results? Does each LF predict a similar or non-similar set of classes? Does human knowledge required to define the specialized class set for each LF?

2. ​In Eq. (7), it is unclear why the potential function should be divided into 4 parts. Citing the paper [14] does not really give any insight about the underlying mechanism. It is suggested to analyze the design principles of this equation carefully.

3. How are different specialized classifiers jointly optimized during training?

4. The motivation of the accuracy estimation and the use of the regularizer in Eq. (13) is unclear. Please explain how the newly introduced loss term (Eq. (13) helps to regularize the model.

5. At test time, how are the final predictions obtained through different specialized classifiers?


**Experiments**

The experiments are well described. However, there are a couple of important comparisons and explanations missing.

1. As the authors mention using multiple data augmentation strategies such as RandomAugment and CutOut. It is unclear how they contribute to the model. It is suggested to also analyze these strategies, and discuss whether the other competitors such as FixMatch and ReMixMatch use the same data augmentation techniques or not to ensure a fair comparison wrt the state-of-the-art.

2. The evaluation metrics change from the main comparison to the ablation study. It is suggested to explain why introducing different new metrics such as macro precision/recall/F1 score and coverage to evaluate the method in Section 4.4. Please explain how these metrics quantify the model performance.

3. How are the number of LFs (K) and the ratio of specialists decided together? In the ablation study, it is indicated that one has to decide the number of LFs first, and the ratio of specialists second. How about the other way around?

4. What do specialists mean? How do specialists define for the ablation study in Table 4?

5. It is suggested to also compare to the recent method [a].

[a] Unsupervised dataaugmentation for consistency training. NeurIPS, 2020

**Time Spent Reviewing:**

3

---

> ### Author Response · Authors · 2021-08-10
> **Response to Reviewer 8Nkc (part 1)**
>
> ####  Method
>
> Q1. **How many LFs are required? How does the randomness in the division of class space affect the results? Does each LF predict a similar or non-similar set of classes? Does human knowledge required to define the specialized class set for each LF?**
>
> Response:
>
> * We use 50 LFs in our experiments.
> * Random initialization of multiple classifiers will affect the division of class space. However, the constraint in Eq.(2) restricts the overall class space distribution, where a sample from any category is trained with $\rho \times K$​​ LFs.
> * A small proportion of  LFs may predict the same specialized set, while others generate the non-similar sets. Take one trained model on CIFAR-10 with 40 labels as an example, there are 37 distinct specialized sets among 50 LFs. Some LFs only specialize in one class but abstain from all other classes. Some specialized sets contain two or more specialized categories. More details of the specialized set analysis will be added to the revised appendix. Trivial visual examples of specialized sets can be referred to Fig.1 a) and b) in [2] and Fig. 3 b), c) and d) in [3].
> * No. Recall that we set $u_k=1$​​​​​​ in Eq. (2) when $H(y_l, \tilde{p}_l^{w,k})$​​​​​​ is the top-$(\rho\times K)$​​​​​​​​​​​ small loss over K LFs. The assignment is automatically done.
>
> Q2. **Why the potential function should be divided into 4 parts?**
>
> Response: First, we extend the dimension of parameters $\theta$ in the label model to $C\times K$, because $\theta \in \mathbb{R}^K$ in Sec. 2.2 in [1] is not suitable for our work as each LF in our paper specializes in multiple categories, and the accuracy of different categories in one LF is different.
>
> Intuitively, the basic idea is to set the potential function as $\phi(y,\tilde{{y}}^k)=exp(\theta_{ky})$. In this case, $P(y, \tilde{\textbf{y}})=\frac{1}{Z}\prod_{k=1}^{K}exp(\theta_{ky})$. We can see that $P(y, \tilde{\mathbf{y}})$ relies  only on $y$, and has no relationship with $\tilde{\mathbf{y}}$. Meanwhile, if we set $\phi(y, \tilde{y}^k)=exp(\theta_{k\tilde{y}})$, there would be no relationship between the final prediction $y$ of label model and  the predicted probability.
>
> Then, we update the potential function  as $\phi(y,\tilde{y}^k)=\begin{cases}	e_{ky}, &if \\; \tilde{y}^k \in \tau_k\\\\ 1. &otherwise
> \end{cases}$, where $e_{ky}:=exp(\theta_{ky})$. It can be observed that the potential function of one LF is triggered only when the LF predicts a specialized category. However, such potential function is sensitive to initilization. We will get 100% accuracy when the parameter for the agreeing case: $\phi(y=\tilde{y}^k, \tilde{y}^k)$ is larger than $\phi(y\neq\tilde{y}^k, \tilde{y}^k)$, and 0% accuracy when $\phi(y=\tilde{y}^k, \tilde{y}^k)$ is smaller. Thus, we have to carefully initialize the parameters $\theta$ so that $\theta_{k\tilde{y}}$ does have larger values.
>
> Next, it is natural to write the potential function as $\phi(y,\tilde{y}^k)=\begin{cases}
> 		1 + e_{ky}, &{ if \\;\tilde{y}^k\in \tau_k,\\; \tilde{y}^k=y}\\\\
> 		1/(1+e_{ky}), &if \\; \tilde{y}^k\in \tau_k,\\; \tilde{y}^k \neq y\\\\ 1. &otherwise\\\\
> 	\end{cases}$,
>
> where we impose a positive impact on the final prediction when $\tilde{y}^k$ agrees with the target $y$ and a negative impact when $\tilde{y}^k$ conflicts with the target $y$. However, the conclusion from [2, 3] tells us that MCL tends to be overconfident for the samples whose ground-truth labels are out of the specialized set. In other words, when one LF is fed with a sample whose ground-truth label is out of the specialized set, it may still produce a label in the specialized set with high confidence. Although such cases decreases a lot due to the introduction of the "abstention" category in Step 1, these cases still exist in our framework. Therefore, we introduce the item of $e_{ky}$ to represent the relationship between the predicted label $\tilde{y}^k$ of the $k_{th}$ LF and the target label $y$, even when the target labels conflict with the predicted labels from LF.
>
> Based on these considerations, we define the four-part potential function as $\phi(y,\tilde{y}^k)=\begin{cases}
> 		1 + e_{ky}, &{ if \\; y \in \tau_k,\\;\tilde{y}^k\in \tau_k,\\; \tilde{y}^k=y}\\\\
> 		1/(1+e_{ky}), &if \\; y \in \tau_k,\\;\tilde{y}^k\in \tau_k,\\; \tilde{y}^k \neq y\\\\
> 		e_{ky}, &if \\; y \notin \tau_k, \\; \tilde{y}^k \in \tau_k,\\; \tilde{y}^k \neq y \\\\ 1. &otherwise
> 	\end{cases}$.
>
> Q3. **How are different specialized classifiers jointly optimized during training?**
>
> Response: All LFs in our method are trained jointly. Concretely, as claimed in L.175-178, we first train LFs jointly with $\mathcal{L}_1(x_l^w,y_l)$ to obtain the specialized set automatically, then use the other two losses $\mathcal{L}_2$ and $\mathcal{L}_u$ to optimize these classifiers in a semi-supervised manner.
>
> In $\mathcal{L}_1(x_l^w, y_l)$​​​​​​, we only supervise the LFs with top-$(\rho \times K)$​​​​​​ small loss for each instance and ignore the cross-entropy of other LFs. For a clear presentation of  $\mathcal{L}_1(x_l^w,y_l)$​​​​​​, we take $K=4$​​​​​​, $\rho=0.5$​​​​​​ for example. Suppose that two labeled samples $x_1$​​​​​​ and $x_2$​​​​​​, the corresponding cross-entropy of 4 LFs are [0.1, 0.4, 3.1, 5.1] and [0.4, 0.1, 0.2, 2.1], respectively. Then, the indicator variable $u$​​​​​​ is derived as [1, 1, 0, 0] and [0, 1, 1, 0] via top-$(\rho\times K)$​​​​​​ small operator. Afterwards, LFs with top-$(\rho\times K)$​​​​​​ small losses are optimized. That is,  $LF_1$​ and $LF_2$​ are optimized with corresponding cross-entropy of $x_1$​, $LF_2$​ and $LF_3$​ are optimized with corresponding cross-entropy of $x_2$​​​​​​​.
>
> After obtaining the specialized set, we use $\mathcal{L}\_2$​ and $\mathcal{L}\_u$​ to train multiple classifiers. For better understanding, we take $K=4$​, $\rho=0.5$​, $|\mathcal{Y}|=3$​ for example. Assume that $\tau_1=\{1\}$​, $\tau_2=\{1, 2\}$​, $\tau_3=\{2,3\}$​, $\tau_4=\{3\}$​. When we are fed with a labeled sample $x_{l}$​ with ground-truth label $y_{l}=2$​, $\mathcal{L}\_2(x\_{l}^w, y\_{l})=H\_1(y=0, \tilde{q}\_l^{w,1}) + H\_2(y=2, \tilde{q}\_l^{w,2})+H_3(y=2, \tilde{q}\_l^{w,3}) + H\_4(y=0, \tilde{q}\_l^{w,4})$​, where $H\_k$​ denotes the cross-entropy of the $k$-th​ LF . Similarly, as for the unlabeled samples $x\_u$​, suppose the weakly augmented probability $\tilde{q}\_u^{w,k}$​ of 4 LFs are $\tilde{q}\_u^{w,1}$​= [0.9, 0.1, 0, 0], $\tilde{q}\_u^{w,2}$​=[0.2, 0,1, 0.7, 0], $\tilde{q}\_u^{w,3}$​=[0.02, 0, 0.97, 0.01], and $\tilde{q}\_u^{w,4}$​=[0.99, 0, 0, 0.01]. Note that first item in each $\tilde{q}\_u^{w,k}$​ represents the probability of abstention. Then, $\mathcal{L}(x\_u^w, x\_u^s)=0 + 0 + 1\*H\_3(y=2, \tilde{q}\_u^{s,3}) + 1\*H\_4(y=0, \tilde{q}\_u^{s,4})$​​​​​​​​​​.
>
>
> Q4.  **Why Eq. (13) helps to regularize the model?**
>
> Response: Consider a binary classification task and a set $S$ of LFs specialize in discriminating class A but not class B. Assume that all unlabeled samples with ground-truth class B mis-trigger one or more LFs in $S$. Even when our four-part potential function is utilized, as training progresses, the likelihood of unlabeled samples (the second term in Eq. (9)) will still be globally maximized when all LFs favor the same class on all instances. Therefore,  the regularizer of incorporating the global accuracy into the trained model can help to avoid such instability and make the model robust.
>
> Q5. **How are the final predictions obtained through different specialized classifiers at test time?**
>
> Response: In the inference phase, we only utilize the end model, which is also a single model and follows the same architecture as FixMatch.
>
> Q6. **It is unclear why it is better than existing methods like FixMatch.**
>
> Response: Better annotation performance of unlabeled data are the key point for our method to outperform the existing methods.
>
> First, we utilize multiple semi-supervised classifiers with the MCL mechanism to obtain the noisy predictions. However, directly integrating the noisy labels still leads to a bad annotation performance, as Majority Vote does in Tab. 3. The averaging results would be similar because the number of specialized LFs for each category is different (as shown in Fig. 3 in our Appendix).
>
> Then, we propose a label model to infer the probabilistic label from the noisy prediction. As shown in Tab.3, our label model on unlabeled samples achieves a much better annotation performance. As mentioned in L.278-281, the label model with 40 labeled samples outperforms the proxy label generator in FixMatch. Thus, our end model learned from a better probabilistic label achieves a better error rate.

---

> > ### Author Response · Authors · 2021-08-10
> > **Response to Reviewer 8Nkc (part 2)**
> >
> > #### Experiments
> >
> > Q7. **It is unclear how multiple data augmentation strategies contribute to the model. It is suggested to also analyze these strategies, and discuss whether the other competitors such as FixMatch and ReMixMatch use the same data augmentation techniques or not to ensure a fair comparison wrt the state-of-the-art.**
> >
> > Response: As we describe in Sec. 4.1, our method and the other competitors such as FixMatch and USADTM have similar data augmentation techniques and network architecture, which ensure a fair comparison wrt the state of the art.
> >
> >
> > Q8. **Explain why introducing different new metrics such as macro precision/recall/F1 score and coverage to evaluate the method in Sec. 4.4. How these metrics quantify the model performance.**
> >
> > Response: We introduce the new metrics to evaluate the performance of produced probabilistic labels, which are also utilized as the hard pseudo labels in other methods. Concretely, the performance of probabilistic label is denoted as annotation performance in Sec. 4.4 of our paper, which is the most critical intermediate result for explaining the great improvement of our methods on a few annotated samples. Briefly, better annotation performance usually corresponds to better final accuracy.
> >
> > Q9. **How are the number of LFs and the ratio of specialists decided together?**
> >
> > Response: The number of LFs and the ratio of specialists are decided through a grid search of the performance on CIFAR-10 with 40 labels. In our experiments, these two hyper-parameters are searched from $\rho \times K := \{0.1, 0.2, 0.3, 0.5\} \times \{10, 20, 40, 50, 60\}$. Detailed annotation performances are listed in the following table.
> >
> > |      |  0.1  |  0.2  |  0.3  |  0.4  |  0.5  |  1.0  |
> > | :--: | :---: | :---: | :---: | :---: | :---: | :---: |
> > |  10  | 83.25 | 86.42 | 88.99 | 90.62 | 91.28 | 92.39 |
> > |  20  | 89.16 | 91.27 | 91.83 | 92.31 | 92.52 | 92.48 |
> > |  30  | 91.39 | 93.46 | 93.54 | 93.16 | 92.91 | 92.41 |
> > |  40  | 92.53 | 93.28 | 93.45 | 93.24 | 92.80 | 92.47 |
> > |  50  | 92.95 | 93.82 | 93.55 | 93.19 | 93.07 | 92.46 |
> > |  60  | 93.01 | 93.54 | 93.11 | 92.97 | 92.63 | 92.43 |
> >
> > Q10. **What do specialists mean? How do specialists define for the ablation study in Tab. 4?**
> >
> > Response: Recall that in Eq. (2), we select classifiers with top-$(\rho\times K)$ small loss to be optimized. As shown in Fig. 1 a) and b) in [2] and Fig. 3 b), c) and d) in [3], the high concentration of each column indicates the specialization of the corresponding model. For better understanding, we present the class-wise accuracy results of some LFs in our model in the following table. We can see that these LFs achieve high accuracy for some specific categories. In our paper, those classes with high accuracy are defined as the specialized classes of an LF, and such an LF with specialized classes is called a specialist.
> >
> > |            |   $LF_1$    |        $LF_2$         |      $LF_3$       |         $LF_4$         |
> > | :--------: | :---------: | :-------------------: | :---------------: | :--------------------: |
> > |  Airplane  |      0      |         0.970         |         0         |           0            |
> > | Automobile |      0      |           0           |         0         |           0            |
> > |    Bird    |      0      |           0           |         0         |         0.925          |
> > |    Cat     |      0      |           0           |         0         |           0            |
> > |    Deer    |      0      |           0           |         0         |           0            |
> > |    Dog     |      0      |           0           |         0         |         0.941          |
> > |    Frog    |      0      |           0           |       0.991       |           0            |
> > |   House    |    0.977    |           0           |       0.981       |           0            |
> > |    Ship    |      0      |           0           |         0         |           0            |
> > |   Truck    |      0      |         0.972         |         0         |         0.977          |
> > |  $\tau_k$​  | $\lbrace House \rbrace$ | $\lbrace Airplane, Truck \rbrace$​ | $\lbrace Frog, House \rbrace$ | $\lbrace Bird, Dog, Truck \rbrace$ |
> >
> > Q11. **It is suggested to compare the recent method [4].**
> >
> > Response: Actually, we compared our method with [4], which is abbreviated as UDA in Tab. 1.
> >
> >
> >
> > [1] Ratner, Alexander, et al. "Snorkel: Rapid training data creation with weak supervision." The VLDB Journal 29.2 (2020): 709-730.
> >
> > [2] Lee, Kimin, et al. "Confident multiple choice learning." International Conference on Machine Learning. PMLR, 2017.
> >
> > [3] Tian, Kai, et al. "Versatile multiple choice learning and its application to vision computing." Proceedings of the IEEE/CVF Conference on Computer Vision and Pattern Recognition. 2019.
> >
> > [4] Unsupervised data augmentation for consistency training. NeurIPS, 2020

---

### Official Review · Reviewer_YU4M · 2021-07-16

**Rating:** 6
**Confidence:** 2

**Summary:**

The paper proposes DP-SSL to address semi-supervised learning by generating probabilistic labels for unlabeled samples with a newly designed data programming (DP), multiple-choice learning (MCL).

The proposed algorithm requires no initial labeling functions (LFs) from human experts and generates LFs from scratch automatically.

Empirical results show that the proposed algorithm achieves higher accuracy with less labeled data on several semi-supervised learning benchmarks.

**Limitations And Societal Impact:**

No obvious social impact.

**Main Review:**

Strength:

The paper relaxes the requirement of human experts to provide initial LFs.

The paper is well organized with a fluent clarification of the approach. I'm not an expert in the semi-supervised learning area but can still understand the paper as a general machine learning researcher.

The paper compares the proposed approach with the state-of-the-art semi-supervised learning approaches, which shows the convincing performance of the proposed approach.

Constraint:

Equation (3) also minimizes the classification error between the ground-truth label and prediction, which may include the function of Equation (2). Could authors explain why Equation (2) is still needed in Equation (5) given Equation (3).

The paper requires multiple stages to train the model, which may need much time to converge. I recommend the authors to draw the training curve for the paper.

Minor points:

Line 89. '.' before 'which' should be ','.

**Time Spent Reviewing:**

2

---

> ### Author Response · Authors · 2021-08-10
> **Response to Reviewer YU4M**
>
> Q1: **Why Eq. (2) is still needed in Eq. (5) given Eq. (3)?**
>
> Response: Actually, loss in Eq. (2) with constraints in L.158 can be reformulated as: we only supervise the LFs with top-$(\rho \times K)$​​ small loss for each instance and abstain the cross-entropy of other LFs.
>
> On the one hand, we need to learn the specialized set of each LF via Eq. (2). After that, we employ the SSL strategy based on the obtained specialized set in Eq. (3) and Eq. (4).
>
> On the other hand, in Eq. (2), only top-$(\rho \times K)$​  small losses are kept as the supervision signals, losses in other LFs are ignored. While in Eq. (3), samples not specialized by one LF are supervised as an abstention category, which affects the back-propagation process.
>
> C1: **Recommend to draw the training curve for the paper.**
>
> Response:  Thanks for your suggestion. We will add the training curve to the revised manuscript.

---

### Official Review · Reviewer_AJRp · 2021-07-27

**Rating:** 6
**Confidence:** 3

**Summary:**

This paper proposed to use multiple-choice learning to generate label functions and add an abstention class for each label function. Then the soft labels are inferred and used to train the final semi-supervised model. Extensive experiments on typical semi-supervised benchmarks show improvements over FixMatch.

**Limitations And Societal Impact:**

The limitations are discussed. There is no negative societal impact.

**Main Review:**

The proposed method first transforms the features then uses different labeling functions to get predictions, which are aggregated to obtain the final soft label. The final model is trained with the supervised loss and the cross-entropy loss between the predictions and the soft labels for the unlabeled data. Experiments are performed on typical SSL benchmarks, including CIFAR10, CIFAR100, SVHN, and STL. The proposed method is often the best, especially when there are fewer labels available. Ablation studies are also provided. Overall, the proposed method is reasonable and works well.

Questions:
- Another baseline is to use an ensemble model plus self-training. I'm not sure if this is exactly the same as the Exp1 in Table 4, but you can average the predictions from different supervised models, then do self-training using these soft labels.
- In the ablation studies, the ensemble baseline (Exp1) is already better than FixMatch, so is the benefit mostly coming from the feature transformation part?
- Methods such as FixMatch only train one single model, but the proposed method requires two-stage training, where the first stage is to train labeling functions. Maybe compare with other self-training methods?
- Does the method also work on datasets with larger image sizes such as ImageNet?

**Time Spent Reviewing:**

4

---

> ### Author Response · Authors · 2021-08-10
> **Response to Reviewer AJRp**
>
> Q1. **Another baseline to use an ensemble model plus self-training with soft labels. I'm not sure if this is exactly the same as the Exp1 in Table 4, but you can average the predictions from different supervised models, then do self-training using these soft labels.**
>
> Response: Thank you for the suggestion. No, it is not the same as Exp1. In Tab. 4, Exp1 without MCL means that we first train multiple semi-supervised classifiers independently, then integrate the predictions of these classifiers with the label model. It can be regarded as multiple semi-supervised classifiers plus a label model. Moreover, another baseline, named Majority Vote in Tab. 3, integrates the noisy predictions and achieves a much lower annotation performance. Concretely, Majority Vote in Tab. 3 can be regarded as multiple semi-supervised classifiers with MCL plus basic voting ensemble strategy. The performance of Majority voting is much lower than ours, which means that such high annotation performance should be attributed to the additional label model rather than the ensemble strategy.
>
> Moreover, directly averaging the predictions from multiple classifiers and doing self-training with these soft labels result in worse accuracy, which varies from $83.95$%​ to $89.43$%​​​​ on CIFAR-10 with 40 labels.​ ​​ ​
>
>
>
> Q2. **Comparing with FixMatch, is the benefit of Exp1 mostly coming from the feature transformation part?**
>
> Response: No. Feature transformation is equipped for MCL. There is no feature transformation part in Exp1. In fact, Exp 1 in Tab. 4 means multiple semi-supervised classifiers with a shared backbone, plus a Label Model, which can be regarded as multiple FixMatch plus a label model. Meanwhile, the annotation performance $90.42$%​​ of averaging multiple classifiers on CIFAR-10 with 40 labels further verifies that the large improvement does not come from the ensemble strategy but the label model.
>
>
>
> Q3. **FixMatch only trains on a train single model. Maybe compare with other self-training methods?**
>
> Response: Thanks for your comment.  We don't think there needs comparison with other self-training methods. First, FixMatch is one SOTA method of semi-supervised image classification. In the training phase, it provides the pseudo labels from weakly-augmented data for the strongly augmented data. So it also can be regarded as a self-training method. The same is true for USADTM. Second, in the inference phase, our method utilizes only the end model, which is also a single model and follows the same architecture as FixMatch.
>
>
>
> Q4. **Does the method also work on datasets with larger sizes such as ImageNet?**
>
> Response: Most existing methods have not been tested on ImageNet. Actually, only FixMatch was tested on ImageNet. So for a fair comparison, we present only comparison results on four widely-used datasets: CIFAR-10, CIFAR-100, SVHN, and STL-10. We believe that our model is consistently capable of boosting the performance on large-scale datasets such as ImangeNet, and we will add such results in the revised version.

---

> > ### Comment · Reviewer_AJRp · 2021-09-03
> > **Thanks for your response**
> >
> > Thank you for clarifying the experimental settings, and providing more results on averaging the predictions. The proposed method indeed provide significant performance gains on CIFAR/SVHN/STL-10. I still think that since the proposed method requires two-stage training, and FixMatch only performs one-stage training, maybe it's good to compare with self-training methods such as [1]. It would also be more useful if authors could experiment on more realistic datasets.
> >
> > [1] Chen et al., Big Self-Supervised Models are Strong Semi-Supervised Learners., NeurIPS 2020.

---

> > > ### Author Response · Authors · 2021-09-03
> > > **Response to Reviewer AJRp**
> > >
> > > Q1: **good to compare with self-training methods such as [1].**
> > >
> > > Response: Thanks for your comment. For ReMixMatch, Fixmatch, USADTM and our method, all these methods outperform [1] with 250 labeled samples in CIFAR-10 as shown in Figure G.1. Actually, [1] benefits a lot from the self-supervised and fine-tune techniques. Thus, the performance on small datasets or datasets with a few labeled data is limited and even worse than those of semi-supervised methods.
> > >
> > >
> > > Q2: **It would also be more useful if authors could experiment on more realistic datasets.**
> > >
> > > Response: Thanks for your suggestion. For a fair comparison, we present only comparison results on four widely-used datasets in semi-supervised methods: CIFAR-10, CIFAR-100, SVHN, and STL-10. We will try to conduct experiments on more realistic datasets in the future.
> > >
> > >
> > > [1] Chen et al., Big Self-Supervised Models are Strong Semi-Supervised Learners., NeurIPS 2020.

---

### Decision · Program_Chairs · 2021-09-28

**Decision:**

Accept (Poster)

**Comment:**

The paper proposes an approach to semi-supervised learning that trains a number of classification heads with different subset of classes, as well as a label model that aggregates the prediction of these classification heads. This approach seems to work well in a self-training setting especially when only few labeled examples are available. Overall, the paper is successful in building a very competitive system for semi-supervised learning (SSL) and it achieves SOTA on several SSL benchmarks. That said, the method has many components and hyper-parameters and it is not very clear where exactly the gains come from. Nevertheless, all of the reviewers have rated the paper slightly above acceptance bar after useful feedback from the authors, and all reviewers are happy to see the paper published. One of the paper's contributions is uniting existing work on data programming and SSL, and I believe, the ideas discussed in this paper can benefit the research community and help develop even better and simpler SSL algorithms. Hence, I recommend acceptance as a poster. Many comments about clarity and ablations have been raised by reviewers. I encourage the authors to address those comments in the camera ready version.

**Consistency Experiment:**

NeurIPS has a long history of experimentation. In 2014, NeurIPS ran an experiment in which 10% of submissions were reviewed by two independent committees to quantify the randomness in the review process. This year, we repeated a variant of this experiment to see how the quality of the review process has changed over time.  This paper was part of the experiment and was therefore assigned to two committees (consisting of reviewers, an Area Chair, and a Senior Area Chair) that reached independent decisions.  If both committees made the same recommendation, this recommendation was followed. If a single committee recommended acceptance, the paper was accepted (with the exception of a few cases in which the other committee identified what we considered a fatal flaw, e.g., an error in a key result).

This copy’s committee reached the following decision: **Accept (Poster)**

The other committee assigned to the paper recommended **Reject**.  You can find the other set of reviews, along with any follow up discussion with the authors here:
https://openreview.net/forum?id=XXhXR2SVse